# 4E Interacting Protein as a Potential Novel Drug Target for Nucleoside Analogues in *Trypanosoma brucei*

**DOI:** 10.3390/microorganisms9040826

**Published:** 2021-04-13

**Authors:** Dorien Mabille, Camila Cardoso Santos, Rik Hendrickx, Mathieu Claes, Peter Takac, Christine Clayton, Sarah Hendrickx, Fabian Hulpia, Louis Maes, Serge Van Calenbergh, Guy Caljon

**Affiliations:** 1Laboratory of Microbiology, Parasitology and Hygiene (LMPH), University of Antwerp, 2610 Wilrijk, Belgium; Dorien.Mabille@uantwerpen.be (D.M.); perutha@gmail.com (C.C.S.); Rik.Hendrickx@uantwerpen.be (R.H.); Mathieu.Claes@uantwerpen.be (M.C.); Sarah.Hendrickx@uantwerpen.be (S.H.); Louis.Maes@uantwerpen.be (L.M.); 2Laboratório de Biologia Celular (LBC), Instituto Oswaldo Cruz (IOC/Fiocruz), Rio de Janeiro 21040-900, Brazil; 3Institute of Zoology, Slovak Academy of Sciences, 84506 Bratislava, Slovakia; peter.takac@savba.sk; 4Scientica, Ltd., 83106 Bratislava, Slovakia; 5DKFZ-ZMBH Alliance, Zentrum für Molekulare Biologie der Universität Heidelberg, 69120 Heidelberg, Germany; cclayton@zmbh.uni-heidelberg.de; 6Laboratory for Medicinal Chemistry, Campus Heymans, Ghent University, 9000 Gent, Belgium; fabian.hulpia@gmail.com (F.H.); serge.vancalenbergh@ugent.be (S.V.C.)

**Keywords:** trypanosomiasis, nucleoside analogues, RNAi, drug target, 4E-interacting protein

## Abstract

Human African trypanosomiasis is a neglected parasitic disease for which the current treatment options are quite limited. Trypanosomes are not able to synthesize purines de novo and thus solely depend on purine salvage from the host environment. This characteristic makes players of the purine salvage pathway putative drug targets. The activity of known nucleoside analogues such as tubercidin and cordycepin led to the development of a series of C7-substituted nucleoside analogues. Here, we use RNA interference (RNAi) libraries to gain insight into the mode-of-action of these novel nucleoside analogues. Whole-genome RNAi screening revealed the involvement of adenosine kinase and 4E interacting protein into the mode-of-action of certain antitrypanosomal nucleoside analogues. Using RNAi lines and gene-deficient parasites, 4E interacting protein was found to be essential for parasite growth and infectivity in the vertebrate host. The essential nature of this gene product and involvement in the activity of certain nucleoside analogues indicates that it represents a potential novel drug target.

## 1. Introduction

Human African Trypanosomiasis (HAT), often referred to as sleeping sickness, is a neglected tropical disease prevalent in the African continent in an area delineated by the Sahara, Kalahari, and Namibian deserts [1]. The disease is caused by the human-infective *Trypanosoma brucei gambiense* and *T. b. rhodesiense*, which are transmitted by tsetse flies [2]. The disease is characterized first by a haemolymphatic stage in which patients present general flu-like symptoms such as fever, weight loss, and muscles aches. A meningo-encephalitic stage subsequently develops when parasites have entered the central nervous system and cause severe neurological symptoms and an altered sleep/wake cycle that is characteristic of sleeping sickness [3].

Currently, available therapies are rather limited and have multiple shortcomings. For example, pentamidine, suramin, melarsoprol, and nifurtimox/eflornithine combination therapy (NECT) require intravenous or intramuscular injections requiring patient hospitalization [4]. Additionally, these drugs exhibit problems of toxicity, with melarsoprol being the most toxic, causing reactive encephalopathy in up to 5 per cent of stage-II HAT patients [5]. Recent encouraging developments in HAT treatment were the clinical trial results for fexinidazole as the first oral monotherapy for g-HAT [6]. Fexinidazole has since then been approved by the European Medicines Agency (EMA) and endorsed in the Democratic Republic of Congo (DRC) for the treatment of both stages of HAT, although NECT remains the treatment of choice for severely ill patients [7]. Nonetheless, for all antitrypanosomal drugs, cases of drug resistance have already emerged in the field or were selected experimentally, highlighting a continuous need for novel treatment strategies [8,9,10].

Given that the parasite lacks de novo purine synthesis and thus solely depends on purine salvage [11], interaction with the involved pathways represents a valid strategy to explore novel antitrypanosomal drugs. Nucleoside analogues such as tubercidin and cordycepin were already proven active against *T. brucei* [12,13,14,15]. Tubercidin (7-deazaadenosine), a nucleoside analogue produced by *Streptomyces tubercidicus*, exhibits potent activity against a range of parasites [16,17,18], but is unfortunately extremely toxic to mammalian cells and therefore of limited therapeutic value [14]. The potent activity of tubercidin (**1**) against *T. brucei* inspired Hulpia et al. to explore a range of C7-substituted tubercidin analogues (Figure 1). Substitution of 7-deazaadenosine with electron-poor phenyl groups gave rise to a series of compounds showing micromolar activity. Replacement of the phenyl group by a pyridine ring resulted in **5** exhibiting sub-micromolar antitrypanosomal activity and largely reduced cytotoxicity compared to tubercidin [19]. A combination of structural elements of tubercidin and cordycepin resulted in the development of 3′-deoxy-7-deazaadenosine nucleosides, with 3′-deoxytubercidin (**2**) as the most promising candidate for late-stage sleeping sickness [20]. Likewise, the screening of a series of 7-substituted 7-deazainosine derivatives provided several 6-O-alkylated analogues (**6**–**9**) with nanomolar in vitro potency and is thus interesting for further in vivo evaluation [21]. Another advantage of nucleoside analogues is their high likelihood to cross the blood-brain barrier due to the various purine transporters lining this barrier [22]. Sufficient compound exposure in the central nervous system is essential for efficacy against stage-II of the disease and is a requirement in the current target product profile [23].

For most antitrypanosomal drugs, the mode-of-action (MoA) is not yet fully understood setting hurdles for rational use in the field in view of the possible development of drug resistance. Genome-wide RNA interference (RNAi) screening is a powerful tool for the unbiased identification of genes involved in the MoA of novel compounds [24]. This technique has been applied to the current antitrypanosomal drugs confirming the involvement of known drug targets, such as the amino acid transporter 6 (AAT6) for eflornithine and nitroreductase (NTR) for benznidazole and nifurtimox [25,26,27,28]. RNAi libraries of *Trypanosoma* were already extensively applied in procyclics [29,30,31]; however, bloodstream forms (BSF) are the clinically relevant stage to study the MoA of drugs. Due to their poor transformation efficiency, RNAi libraries in BSF have only emerged later [32,33,34].

In this study, we successfully applied whole-genome RNAi screening in combination with confirmatory in vitro and in vivo approaches to identify genes participating in the MoA of tubercidin analogues that may represent potential novel drug targets given their essentiality for *T. brucei* infectivity.

## 2. Materials and Methods

### 2.1. Ethics Statement

The use of laboratory rodents was carried out in strict accordance with all mandatory guidelines (EU directives, including the Revised Directive 2010/63/EU on the Protection of Animals used for Scientific Purposes that came into force on 1 January 2013, and the declaration of Helsinki in its latest version) and was approved by the Ethical Committee of the University of Antwerp, Belgium (UA-ECD 2017-04).

### 2.2. Animals and Parasites

Female C57BL/6JRj were purchased from Janvier (Le Genest Saint Isle, France). Food for laboratory rodents (Carfil, Arendonk, Belgium) and drinking water were available ad libitum. The animals were kept in quarantine for at least 5 days before infection and randomly allocated to the experimental units.

The RNAi library (kindly gifted by Professor Isabel Roditi, University of Bern) was constructed in the New York “single marker” line, a derivative of MITat 1.2 BSF trypanosomes [27]. Transfected EATRO1125 cells (wild-type (WT), 4E interacting protein double knock-out (4EIP-KO), EIF4E1 double knock-out (4E1-KO), 4EIP add-back (4EIP-AB), and EIF4E1 add-back (4E1-AB) containing a tetracycline-inducible 4EIP- or 4E1-myc gene) [35] were available from the Clayton lab (University of Heidelberg).

### 2.3. Compounds

A range of tubercidin analogues (Figure 1) was synthetized as part of a campaign to find novel hits with enhanced selective potency against *T. brucei* compared to tubercidin [19,20,21]. Experimental details regarding their synthesis are described [19,20,21]. The synthesis of analogue **4** is described in Appendix A. High activity to treat late-stage sleeping sickness was demonstrated for 3′-deoxytubercidin (**2**) [20]. Substitution of C-7 with a bromide, trifluoromethyl, or pyridin-2-yl group led to compounds **3**–**5**, with the latter showing sub-micromolar activity in vitro [19]. Compounds **6**–**9** are 7-substituted 7-deazainosine analogues [21].

### 2.4. In Vitro Cytotoxicity Assay

Human lung fibroblasts (MRC-5_SV2_ cells) were cultured in Minimum Essential Medium (MEM) (Life Technologies, Carlsbad, CA, United States) supplemented with l-glutamine, NaHCO_3_, and 5% heat-inactivated fetal bovine serum (iFBS). Cells were seeded at a concentration of 15,000 cells/well, to which 4-fold dilutions of the test compounds were added with 64 µM as the highest in-test concentration. After 72 h of drug exposure at 37 °C and 5% CO_2_, cell viability was determined by fluorescence reading (Tecan^®^, GENios, Männedorf, Switzerland) after a 4-h incubation with resazurin (Sigma Aldrich, St. Louis, MO, USA). The 50% cytotoxic concentration (CC_50_) was calculated for each of the compounds.

### 2.5. Resazurin-Based Susceptibility Assay

*T. brucei brucei* MITat 1.2 New York “single marker” cells (NY-SM) were cultured in Hirumi’s modified Iscove’s medium 9 (HMI-9) with 10% iFBS. To determine the in vitro susceptibility, NY-SM were seeded at a concentration of 4000 parasites/well. Four-fold dilutions of the test compounds were added with a highest in-test concentration of 64 µM. After 72 h of drug exposure at 37 °C and 5% CO_2_, the viability was determined by fluorescence reading (Tecan^®^, GENios, Männedorf, Switzerland) after a 6-h incubation with resazurin (Sigma Aldrich, St. Louis, MO, USA). The 50% inhibitory concentration (IC_50_) was calculated by comparing cell viability to untreated control wells [36].

### 2.6. RNAi Library Screening

The RNAi library was cultured in 30 mL HMI-9 culture medium supplemented with 10% iFCS and 1 µg/mL tetracycline to induce the RNAi phenotype. Ten mL of culture without tetracycline induction served as a negative control. After 62 h of induction, different concentrations of **5** (400, 600, and 800 nM) were added to 10 mL of individual cultures containing 1 × 10^5^ cells/mL. The compound concentrations were based on their IC_98/99_ values as determined in the resazurin-based susceptibility assay. The cells were monitored daily. As soon as the non-induced RNAi cultures collapsed and the induced cultures grew normally under drug pressure, they were diluted 1:10 in a fresh medium containing tetracycline and drug compound. After 2–3 passages, the cultures were harvested for further analysis (Appendix A).

### 2.7. RNAi Insert Identification

The genomic DNA (gDNA) of the drug-exposed parasites was isolated from 20 mL of a stationary-phase culture using the QIAamp DNA mini kit (QIAGEN, Hilden, Germany). The DNA concentration was determined using nanodrop and diluted to 20 ng/µL gDNA in PCR water. The RNAi inserts were amplified using p2T7_seq (5′-CCGCTCTAGAACTAGTGGA-3′) as forward and p2T7hygPJ4 (5′-GGAAAGCTAGCTTGCATGCCTG-3′) as reverse primer (PCR program: 1 cycle for 1 min at 98 °C, 25 cycles for 1 min at 98 °C, 30 s at 58 °C, 1 min at 72 °C and 1 cycle for 10 min at 72 °C).

PCR products were analyzed on a 1% agarose gel (100 V for 120′) to identify the number and size of amplified inserts. PCR products were purified using the ExoSAP-ITTM PCR Product Cleanup kit (ThermoFisher Scientific, Carlsbad, CA, USA). A nested PCR was performed (PCR program: 1 cycle for 3 min at 94 °C, 30 cycles for 1 min at 94 °C, 1 min at 50 °C, 2 min and 30 s at 72 °C and 1 cycle for 10 min at 72 °C) and PCR products were analyzed on a 1% agarose gel (100 V for 120′). Interesting inserts, only present in the tetracycline clones, were extracted from the gel using the GenEluteTM Gel Extraction Kit (Sigma-Aldrich, St. Louis, MO, USA).

Finally, the PCR products were ligated in the pCR 2.1 vector system (ThermoFisher Scientific, Carlsbad, CA, USA) and transformed into high-efficiency competent *E. coli* (NEB10) and plated onto LB agar with ampicillin. Five colonies per target were selected to perform a colony PCR (PCR program: 1 cycle for 3 min at 94 °C, 30 cycles for 1 min at 94 °C, 1 min at 50 °C, 2 min and 3s at 72 °C and 1 cycle for 10 min at 72 °C) and were subsequently analyzed on a 1% agarose gel (100 V for 120′). Positive clones were selected and sequenced (VIB Genomics Core, Antwerp). The RNAi insert was sequenced bidirectionally with primers p2T7_seq and p2T7linker_rev (5′-AGGGCCAGTGAGGCCTCTAGAG-3′). Sequences were blasted against *T. brucei* transcripts on the TriTrypDB [37]. Retrieved translated sequences were subjected to characteristics determination using the ProtParam [38] and SignalP tool [39] and homology detection using a comparison of hidden Markov Models [40]).

### 2.8. Independent Confirmation of the RNAi Phenotype

To validate the RNAi effect of the different RNAi inserts, the insert was back-cloned into a p2T7Bern vector for transformation into NY-SM cells (Appendix A). The RNAi inserts were introduced together with Ndel and Xhol restriction sites in the PCR 2.1 vector using P2T7_seq forward primer containing the NdeI site (GGAATTCCATATGCCGCTCTAGAACTAGTGGA) and the P2T7linker_rev reverse primer containing the XhoI site (CCGCTCGAGAGGGCCAGTGAGGCCTCTAGAG). PCR products were analyzed on a 1.2% agarose gel and correct lanes were extracted from the gel using the GenEluteTM Gel Extraction Kit (Sigma-Aldrich). The purified PCR product and the p2T7Bern vector were digested overnight at 37 °C with NdeI and XhoI restriction enzymes. The digested DNA was ligated using a T4 ligase for 2 h at room temperature. Finally, the ligated products were transformed into high-efficiency competent *E. coli* (NEB10) by electroporation and plated onto LB agar with ampicillin. A colony PCR was performed to identify the colonies containing the vector followed by confirmation by sequencing.

Next, 10 µg of the P2T7Bern DNA containing the RNAi inserts was linearized with NotI and purified using the QIAquick PCR purification kit (QIAGEN, Hilden, Germany); 4 × 10^7^ NY-SM cells were transferred to a 2 mm gap cuvette together with 10 µg linear plasmid DNA. The cells were subjected to two consecutive pulses of 1.2 kV with 186 Ω resistance and 50 µF capacitance. After transformation, the cells were transferred to HMI-9 medium containing 10% iFCS. After 18–22 h, 2.5 µg/mL of hygromycin B was added to the cells. After 6 days, the hygromycin B pressure was increased to 5 µg/mL. After sufficient cell growth, the culture was used to generate monoclonal cell lines. Monoclonal lines were subjected to resazurin-based susceptibility assays to determine the 50% inhibitory concentration, as described above.

### 2.9. Efficiency of the RNAi Silencing

To determine the degree of knock-down of the different RNAi inserts after tetracycline induction, an RT-qPCR was performed. The different RNAi clones were cultured in HMI-9 culture medium in the presence or absence of 1 µg/mL of tetracycline. Cultures were diluted 1 in 10 every 24 h in fresh HMI-9 with or without tetracycline. After 68 h, RNA was extracted using the RNA blood mini kit (QIAGEN, Hilden, Germany). cDNA was prepared using the Tetro™ cDNA Synthesis Kit (Bioline). RT-qPCR was performed using the SensiFAST™ SYBR^®^ Hi-ROX Kit (Bioline) (PCR program: 1 cycle for 2 min at 95 °C, 40 cycles for 5 s at 95 °C, 20 s at 60 °C, 15 s at 95 °C, 1 min at 60 °C and 15 s at 95 °C). *TERT*, *Actin A*, and *H2B* were used as reference genes [41,42].

### 2.10. In Vitro Growth

The in vitro growth of RNAi clones was compared to the NY-SM background strain. Parasites were seeded at a concentration of 4 × 10^5^ cells/mL in 2 mL of HMI-9 with 10% iFBS with or without the addition of 1 µg/mL tetracycline. Cell growth was monitored by microscopic counting using a Neubauer improved haemocytometer for a period of 9 days. Cells were diluted 1:5 daily.

### 2.11. Cell Cycle Analysis and RNA Quantification

NY-SM cells were exposed to different concentrations of 5 (corresponding to the IC_50_, IC_50_/2, and IC_50_/5) for 24 h. For cell cycle analysis, the cells were harvested and washed with PBS before staining with 5 μg/mL Hoechst 33,342 for 25 min at 37 °C. Cells were analyzed on a MACSQuant flow cytometer (Miltenyi Biotec, Gladbach, Germany) using the FlowJo X software package. For RNA quantification, extracts were made from a normalized number of cells to correct for the impact of 5 on the number of recovered trypanosomes. RNA was extracted using the QIAamp RNA blood mini kit (QIAGEN, Hilden, Germany). Total RNA yields were determined using both Nanodrop 2000 spectrometry and Qubit fluorimetric RNA content analysis. Additionally, specific transcripts in the RNA pool were quantified using quantitative reverse transcription PCR (RT-qPCR) targeting the spliced-leader RNA (SL-RNA), 18S rRNA, and transcripts encoding the telomerase reverse transcriptase (*TERT*) (Appendix A) [41,43].

### 2.12. Western Blot Analysis of PAD1

Tetracycline-induced 4EIP-AB parasites were seeded at a concentration of 5 × 10^5^ cells/mL and exposed to various concentrations of **5** and 3′-deoxytubercidin [20] (corresponding to the IC_50_, IC_50_/2 and IC_50_/5) for 48 h. Cells were resuspended in Laemmli sample buffer (Bio-Rad, Hercules, CA, USA) containing β-mercaptoethanol. Genomic DNA was sheared by sonication and samples were stored at −80 °C until further analysis; 10 µL of protein sample was loaded on 10% Mini-PROTEAN^®^ TGX Stain-Free™ Protein Gel (Bio-Rad) at 100 V. Proteins were transferred to a polyvinylidene difluoride (PVDF) membrane at 100 V. Primary antibodies for PAD1 and EF1ɑ (kindly gifted by Prof. Keith Matthews, University of Oxford) were used at 1:1000 and 1:7000, respectively [44]. For protein detection, anti-rabbit (1:7000) or anti-mouse (1:1000) HRP-conjugated antibodies were used with the Clarity Western ECL Substrate (Bio-Rad) detection system.

### 2.13. In Vivo Infectivity

Twelve female C57BL/6JRj were randomly divided into four groups. Two groups (tetracycline-positive groups) were pre-treated for 48 h with 1 mg/mL of doxycycline and 5% sucrose in the drinking water. Doxycycline exposure was maintained throughout the entire experiment and fresh doxycycline solutions were prepared every 2–3 days. The tetracycline-negative groups received water with 5% sucrose. On day 0, mice were infected intraperitoneally with 5000 parasites. Each group was infected with a different strain: NY-SM + (pre-induced with tetracycline in vitro), NY-SM—(not induced with tetracycline), ADKIN A + (NY-SM cells transfected with RNAi insert Tb927.6.2300 and pre-induced with tetracycline in vitro), and ADKIN A—(NY-SM cells transfected with RNAi insert Tb927.6.2300 and not induced with tetracycline). Parasitaemia and anaemia were determined using a Neubauer improved haemocytometer. To confirm adequate knock-down of the genes of interest, 100 µL of blood was collected at day 4 post-infection (dpi). RNA was extracted using the RNA blood mini kit (QIAGEN) and subjected to RT-qPCR to determine the percentage of gene knock-down.

The role of 4EIP RNAi-mediated knockdown (NY-SM cells transfected with RNAi insert Tb927.9.11050) and knockout (4EIP-KO and 4EIP-AB strains) in in vivo infectivity was evaluated in a similar manner as described above.

### 2.14. Tsetse Fly Infections

*T.b.b. AnTat1.1EPpyRE9* parasites were used to infect teneral tsetse flies (*Glossina morsitans morsitans*) through a parasitized blood meal supplemented with 10 mM reduced l-glutathione. The infected blood meal consisted of a mixture of infected mouse blood (5–7 dpi) and defibrinated horse blood to obtain a final concentration of >10^6^ BSF parasites/mL. Newly emerged tsetse flies were divided into three groups and offered an infected blood meal with or without **5** or 3′-deoxytubercidin at a concentration corresponding to the IC_50_. After the infected blood meal, tsetse flies were fed every 2–3 days with uninfected defibrinated horse blood. At 7 dpi, tsetse flies were dissected to evaluate midgut infection ratios.

### 2.15. Graphs and Statistical Analyses

All graphs and statistical analyses were prepared using GraphPad Prism 7 software.

## 3. Results

### 3.1. 7-Deaza Adenosine Analogues Act Independently of DNA and RNA Synthesis Inhibition

All 7-deaza adenosine analogues **2**–**9** showed sub-micromolar activity against *T. brucei* parasites in vitro. Tubercidin was cytotoxic at low concentrations, as described in literature [45]. Analogues **4** and **9** showed a modest increase in selectivity, whereas the remaining analogues showed a significantly improved selectivity (Table 1). Cell cycle analysis by flow cytometry indicated no major impact on DNA synthesis (Figure 2A–D), suggesting that 5 does not inhibit the parasite’s DNA polymerase. The impact of various concentrations of 5 on RNA synthesis was analyzed next using standard quantification methods on the total RNA extracts and specific RT-qPCR for quantitative detection of *T. brucei* SL-RNA, rRNA, and *TERT* transcripts (Figure 2E–H). The results indicate that **5** does not act as a substrate nor chain terminator for RNA synthesis given the absence of a major impact on the total RNA pool and the levels of specific transcripts produced by RNA polymerase I and II.

### 3.2. Involvement of Adenosine Kinase and 4E-Interacting Protein in the Antitrypanosomal Activity

To determine the MoA of **5**, selection of a genome-wide *T. b. brucei* RNAi library identified four RNAi inserts (Appendix A), most notably FLA1-binding protein (FLA1BP; Tb927.8.4050), endonuclease G (EndoG; Tb928.8.4040), adenosine kinase (ADKIN; Tb927.6.2300), and 4E-interacting protein (4EIP; Tb927.9.11050). To confirm the involvement of the different inserts in the MoA of 5, the RNAi inserts were back-cloned into NY-SM cells. Two independent clones of each RNAi insert were tested for their susceptibility to **5**.

Out of the four knock-down constructs tested, only those silencing ADKIN and 4EIP exhibited an approximately 3-fold decrease in susceptibility to **5** (Figure 3). RT-qPCR indicated variable gene silencing levels in the various RNAi clones, suggesting that residual target activity may be responsible for the partial activity of the compound (Table 2).

For the RNAi clones FLA1BP and EndoG, no significant differences in susceptibility could be observed between tetracycline-induced and non-induced clones. To confirm these findings, the in vitro susceptibility assays were also conducted under continuous tetracycline pressure, yielding similar results.

### 3.3. 7-Deaza Adenosine Analogues Show a Variable Dependence on ADKIN and 4EIP

The susceptibility of ADKIN RNAi-mediated knockdown to other 7-deaza purine analogues was evaluated in vitro. Four of the compounds showed a large decrease in susceptibility (Figure 4) of which two (6 and 7) even completely lost their activity (up to the highest concentration tested (64 µM)) against the ADKIN-knockdown strains. The large dependency on ADKIN for antitrypanosomal activity proved particularly relevant for the 6-O-alkylated analogues **6**–**9** (Figure 4, red circles). For 4EIP, the susceptibility to other 7-deaza adenosine analogues was tested using both the RNAi-mediated knockdown and knockout cell lines. Only 5 showed a partial dependence on 4EIP for its antitrypanosomal activity (Figure 5). The effect seemed to be specific for 4EIP and not for the cap-binding protein 4E1. The variable dependence of the different compounds on ADKIN and 4EIP suggests a diverse MoA of the nucleoside analogues.

### 3.4. 4 EIP but Not ADKIN Is Essential for In Vivo Infectivity in Mice

To evaluate the role of ADKIN and 4EIP, the impact of gene silencing on in vitro growth and in vivo infectivity were determined. Knockdown of ADKIN did not alter parasite growth in vitro and parasites retained a normal infectivity profile in mice (Figure 6A,B). In contrast, 4EIP RNAi parasites showed a significant growth deficit already evident after 24 h of tetracycline induction (Figure 6C). In mice, knockdown of 4EIP was responsible for a strikingly reduced infectivity (Figure 6D); however, the mice still succumbed to infection within 2 weeks, most likely due to a loss of the RNAi phenotype in vivo. This hypothesis was confirmed using 4EIP-KO parasites which could be detected during the first week of infection but were undetectable from day 7 onwards until the pre-set endpoint of 70 dpi (Figure 6E). Even after immunosuppressive treatment with 150 mg/kg cyclophosphamide, parasitaemia did not reappear confirming the essential nature of 4EIP for in vivo infectivity.

### 3.5. 5 Does Not Specifically Impact Parasite Differentiation

Given the known involvement of 4EIP in parasite differentiation, the impact of **5** on short stumpy differentiation was assessed. PAD1 protein expression was used as a stumpy-specific marker using Western blot. To discern between the effects of changes in parasite density as a result of the antiparasitic activity and 4EIP-specific compound effects on PAD1 expression, the impact of **5** was compared to that of 3′-deoxytubercidin, which is 4EIP-independent. A concentration-dependent variation in PAD1 expression could be observed, most likely dependent on parasite density instead of target-dependent effects, since the same observations were made for 3′-deoxytubercidin (Figure 7). Additionally, no impact was observed of compound addition to infectious blood meals on midgut infection ratios in the tsetse fly vector (Figure 8).

## 4. Discussion

While limitations of currently used treatments for HAT should stimulate additional drug discovery initiatives, it remains a challenge to identify non-toxic lead compounds capable of crossing the BBB to treat stage-II infection. Since *T. brucei* parasites solely depend on purine salvage, nucleoside analogues may offer opportunities as novel tools against HAT with CNS involvement [11]. Based on the previously identified nucleoside analogues tubercidin and cordycepin with known antitrypanosomal potential, a range of novel analogues were explored. The combination of structural elements of tubercidin and cordycepin resulted in the discovery of 3′-deoxytubercidin as a highly interesting lead compound to treat second-stage sleeping sickness [20]. 7-Deazaadenosines substituted with phenyl groups or a pyridine ring resulted in excellent in vitro potency and selectivity profile against African trypanosomes [19]. Finally, a series of 6-O-alkylated analogues was highly promising with nanomolar in vitro activity against *T. brucei* parasites [21].

The occurrence of drug resistance in the field emphasizes the need for identification of the potential mechanisms of resistance early on in the drug development process. RNAi screening provides a whole-genome unbiased approach for the identification of drug targets [28,46] and has been successfully used in identifying the MoA of the existing antitrypanosomal drugs eflornithine and nifurtimox [26]. In the present study, we used whole-genome RNAi screening to gain insight into the MoA of **5**, whereby the involvement of four targets, namely FLA1BP, EndoG, ADKIN and 4EIP, was demonstrated.

FLA1BP is responsible for joining the flagellar membrane to the cell membrane of the parasite. RNAi knockdown of FLA1BP has been described to cause detachment of the flagellum from the cell body, however, without any impact on parasite proliferation rates [47]. The fact that FLA1BP is more abundantly found in procyclic cells than in BSF may explain the inability to confirm this target and the lack of morphological changes in our FLA1BP knockdown parasites.

EndoG plays a role in trypanolysis induced by the human serum apolipoprotein 1 (APOL1). Uptake of APOL1 by the parasite causes lysosomal and mitochondrial membrane permeabilization and the subsequent translocation of mitochondrial EndoG to the parasite nucleus causing DNA damage and cell death [42]. This target, however, could not be confirmed, possibly due to experimental limitations.

ADKIN is a known enzyme in the purine salvage pathway of the parasite and is responsible for the direct phosphorylation of adenosine to AMP following uptake of the former [48,49]. The reduced sensitivity of ADKIN RNAi strains to **5** suggests the occurrence of compound phosphorylation upon uptake by the parasite. This phenomenon has already been described for other nucleoside analogues [20]. However, the fact that **5** remained active against the ADKIN knockdown strains at sub-micromolar levels indicates that this phosphorylation may not be essential or only partially contributes to the overall activity. It is clear from a broader compound evaluation on the developed ADKIN RNAi strains that various analogues exhibit different degrees of dependency on ADKIN or may be subject to different rates of enzymatic conversion by ADKIN. ADKIN plays an important role in trypanosomal purine salvage [48,49], but the in vitro and in vivo growth curves [50,51] reveal that it is not essential for growth and infectivity. This can be attributed to the fact that loss of ADKIN activity redirects adenosine salvage through the alternative cleavage-dependent pathway. Additionally, the parasite can salvage hypoxanthine, inosine, and adenine to form all of its required purines [48].

The second identified target, 4EIP, has been extensively studied in *Leishmania major*. 4EIP binds with its N-terminal domain to the mRNA cap-binding protein LeishIF4E, leading to dissociation from the cap and a suppression of active translation [52]. Similar binding properties and functions were later described for African trypanosomes [35]. The MoA of **5** can be partially attributed to 4EIP as confirmed by the use of knock-out and add-back control parasites. However, the cap-binding protein 4E1 was not influenced by **5**, indicating that the inhibition of 4EIP works independently of 4E1. The suppression of translation in *T. brucei*, as regulated by 4EIP, is required for successful differentiation into short stumpy parasites in the bloodstream. This is an essential adaptation of the parasite for successful transmission to the tsetse fly vector [35] and subsequent transformation to procyclic parasites in the tsetse fly midgut. In this study, we showed a large growth deficit of 4EIP knockdown parasites in culture conditions, while in mice, 4EIP knockout led to a rapid clearance of parasite burdens. The underlying basis for the essential role of 4EIP in vivo remains to be elucidated and this study could not demonstrate a link between compound activity and inhibition of parasite differentiation in vitro and in the tsetse fly.

In summary, we here describe the acquired insights in the MoA of novel 7-deazaadenosine nucleoside analogues for the treatment of HAT. RNAi screening in combination with confirmatory in vitro and in vivo experiments using knockout strains identified the involvement of 4EIP in the activity of some antitrypanosomal nucleoside analogues. The reliance of some compounds on this gene product and the essential nature for in vivo infectivity indicate that 4EIP represents a potential drug target. The available RNAi and knockout clones will allow a more in depth analysis of the structural requirements for metabolic activation and 4EIP interaction of newly synthetized nucleoside analogues.

## Figures and Tables

**Figure 1 microorganisms-09-00826-f001:**
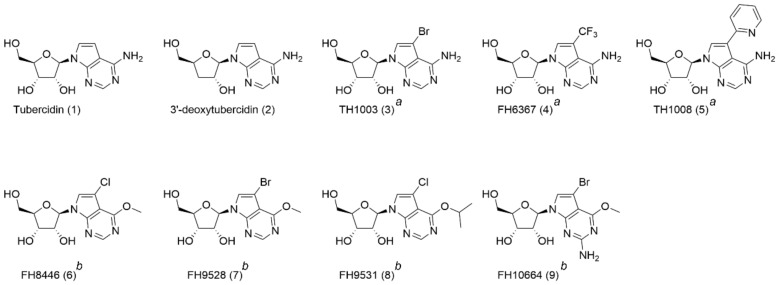
An overview of the nucleoside analogues used in this study. Compound name (compound code). ^a^ 7-deaza adenosine analogues. ^b^ inosine analogues. Compound codes in original publication: (Tubercidin (1), 3′-deoxytubercidin (9)) [20], (TH1003 (31), FH6367, TH1008 (13)) [19], (FH8446 (23), FH9531 (36), FH9528 (24), FH10664 (28)) [21].

**Figure 2 microorganisms-09-00826-f002:**
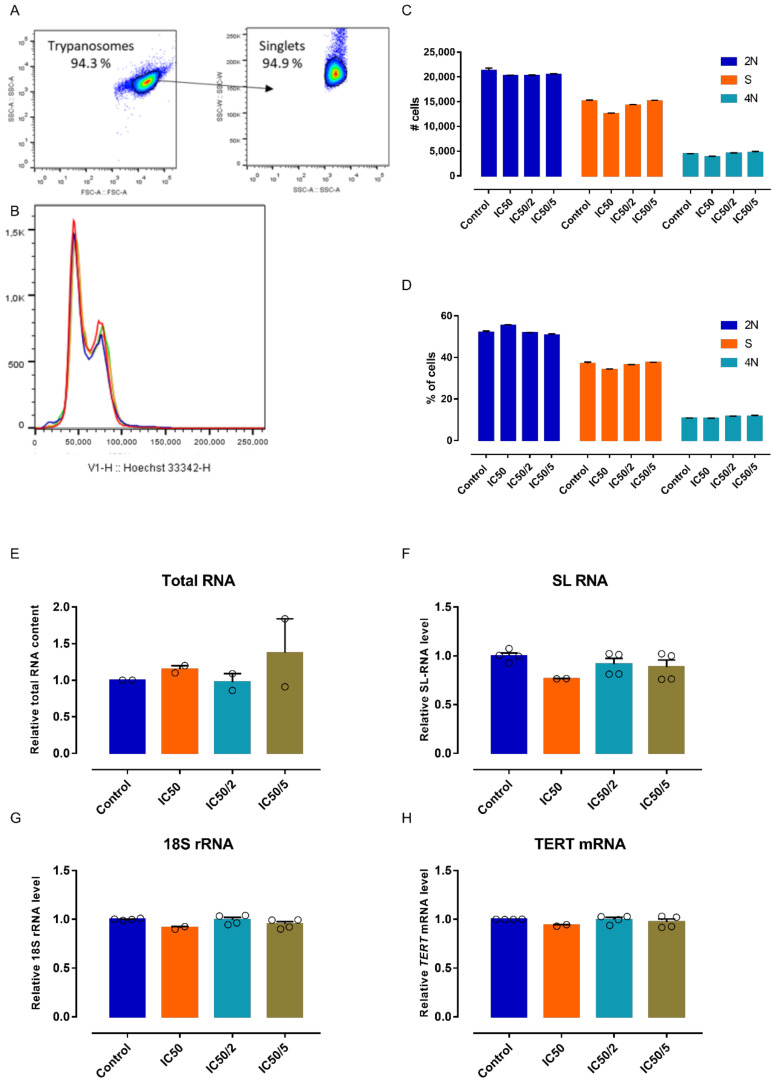
Effect of **5** on cell cycle regulation and RNA synthesis. (**A**) Gating strategy. (**B**) Results of the cell cycle analysis after 24 h exposure to different concentrations of 5. Results of the cell cycle analysis (2N, S phase, 4N) expressed in total cell numbers. (**C**) *n*  =  2 and relative cell numbers. (**D**) *n*  =  2. (**E**) Total RNA yield from trypanosomes exposed for 24 h to different concentrations of 5 (*n*  =  2). RNA content quantified using quantitative reverse transcription PCR (RT-qPCR) specifically targeting SL-RNA. (**F**) *n*  =  2 with two replicates, 18S rRNA. (**G**) *n*  =  2 with two replicates and *TERT* mRNA. (**H**) *n*  =  2 with two replicates. RNA content and transcript levels are expressed as relative to the non-treated trypanosomes (control). Error bars are SEM.

**Figure 3 microorganisms-09-00826-f003:**
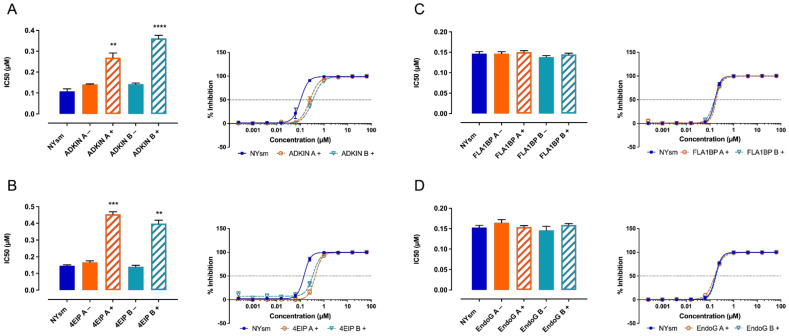
Drug susceptibility of RNAi clones targeting (**A**) ADKIN, (**B**) 4EIP, (**C**) FLA1BP, and (**D**) EndoG to **5**. Results are expressed as the mean IC_50_ (µM), and error bars represent SEM, and are based on at least two independent experiments (*n* = 2), each with two biological replicates. + = tetracycline-induced clones; − = non-induced clones. All experiments were performed with two independently generated RNAi clones. ** *p*  <  0.01, *** *p*  <  0.001, **** *p* < 0.0001 Kruskal–Wallis test with Dunn’s multiple comparison test.

**Figure 4 microorganisms-09-00826-f004:**
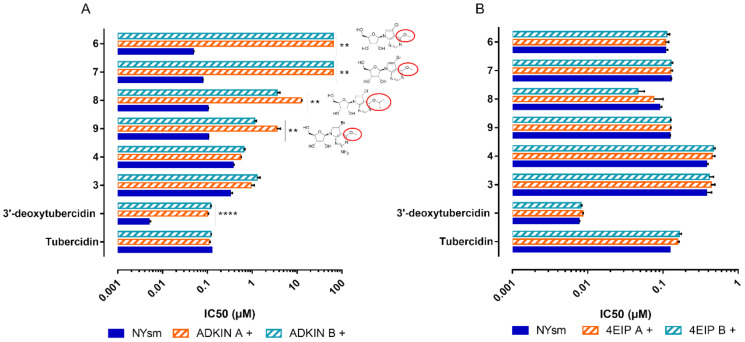
Susceptibility of ADKIN (**A**) and 4EIP (**B**). RNAi-mediated knockdown to compounds **1**–**4**, and **6**–**9**. Results are expressed as the mean IC_50_ (µM), and error bars represent SEM and are based on at least two independent experiments (*n* = 2), each with two biological replicates. + = tetracycline-induced clones. All experiments were performed with two independently generated RNAi clones. ** *p* < 0.01,**** *p* < 0.0001, Mann–Withney test comparing NY-SM with RNAi knock-down strains. ADKIN 3′-deoxytubercidin data was previously published [20].

**Figure 5 microorganisms-09-00826-f005:**
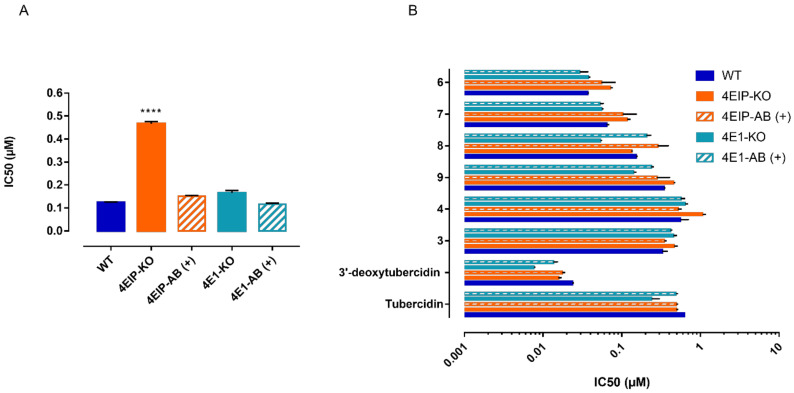
Susceptibility of 4EIP and 4E1 knockout cell lines to **5** (**A**) and compounds **1**–**4**, and **6**–**9** (**B**). Results are expressed as the mean IC_50_ (µM), and error bars represent SEM and are based on at least two independent experiments (*n* = 2), each with two biological replicates. + = tetracycline-induced clones. **** *p*  <  0.0001, Kruskal–Wallis test with Dunn’s multiple comparison test.

**Figure 6 microorganisms-09-00826-f006:**
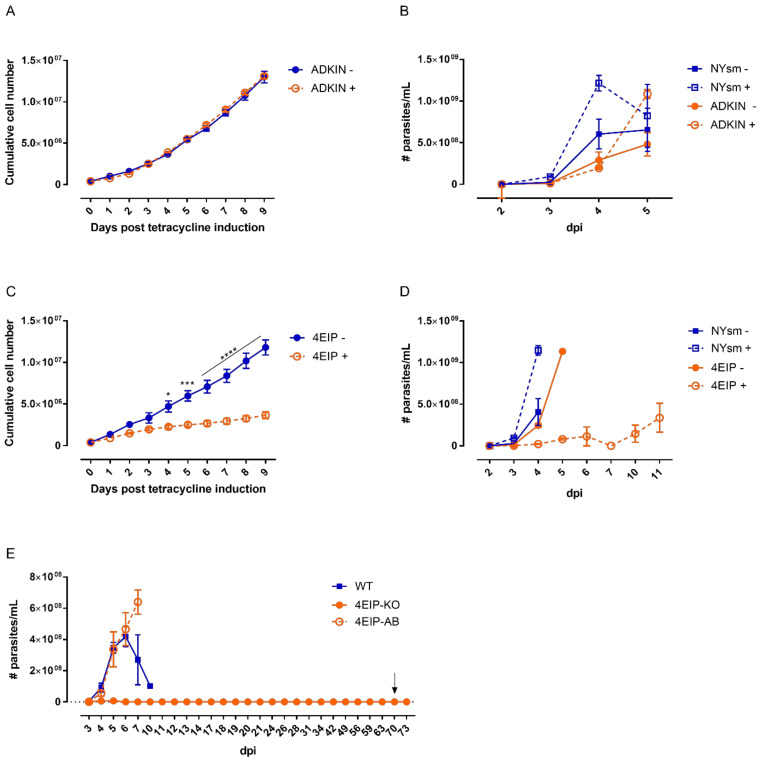
The role of ADKIN and 4EIP in parasite growth and infectivity. Impact of ADKIN RNAi-mediated knockdown on (**A**) in vitro growth and (**B**) and in vivo infectivity. Impact of 4EIP RNAi-mediated knockdown on (**C**) in vitro growth and (**D**) and in vivo infectivity. (**E**) Impact of 4EIP knockout (4EIP-KO) and add-back (4EIP-AB) on in vivo infectivity. The arrow indicates the moment of immunosuppressive treatment. Results are expressed as the average cell numbers, and error bars represent the SEM. The in vitro experiments are based on two independent experiments (*n* = 2), each with two biological replicates. + = tetracycline-induced clones; − = non-induced clones. The in vitro cumulative growth curves were performed with two independently generated RNAi clones. * *p* < 0.01, *** *p* < 0.001, **** *p* < 0.0001.

**Figure 7 microorganisms-09-00826-f007:**
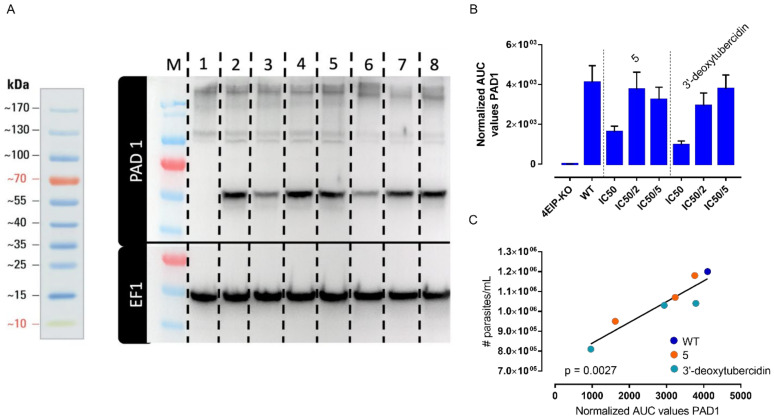
Western blot analysis for PAD1 expression. (**A**) PAD1 and EF1 protein expression (1: 4EIP-knockout, 2: untreated controls, 3–5: **5**-treated with IC_50_, IC_50_/2 and IC_50_/5 and 6–8: 3′-deoxytubercidin-treated with the IC_50_, IC_50_/2 and IC_50_/5). (**B**) EF1-normalized area under the curve (AUC) values for PAD1. (**C**) Correlation between the normalized AUC values for PAD1 and culture density at time of collection.

**Figure 8 microorganisms-09-00826-f008:**
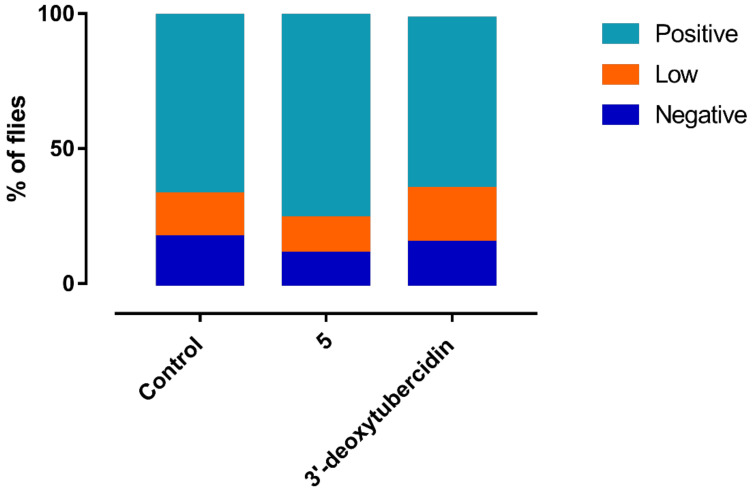
Impact of **5** supplementation on establishment of *T. brucei* in the tsetse fly midgut. Infectious blood meals were supplemented with **5** or 3′-deoxytubercidin at the IC_50_. Results represent the percentage of *G. m. morsitans* flies with a negative, low, or positive infection in the midgut and are a representation of two independent experiments with a minimum of 25 flies per group.

**Table 1 microorganisms-09-00826-t001:** Overview of the in vitro activity and cytotoxicity of the 7-deaza adenosine analogues on NY-SM cells (IC_50_) and MRC-5 cells (CC_50_) [19,20,21].

Compound	IC_50_ ± SEM (µM)	CC_50_ ± SEM (µM)	SI ^1^
Tubercidin (1)	0.12 ± 0.003	2.23 ± 0.68	18
3	0.38 ± 0.064	33.3 ± 12.6	88
4	0.38 ± 0.017	19.2 ± 15.2	50
5	0.15 ± 0.005	15.1 ± 4.1	103
6	0.05 ± 0.003	48 ± 16	960
7	0.08 ± 0.001	>64	800
8	0.10 ± 0.008	>64	640
9	0.10 ± 0.001	3.24 ± 0.01	32

^1^ The selectivity index (SI) represents the CC_50_/IC_50_. SEM: Standard Error of the Mean.

**Table 2 microorganisms-09-00826-t002:** Overview of the RNA knock-down levels following in vitro induction with tetracycline.

	% Knock-Down	
Target	Clone A ^1^	Clone B ^1^
FLABP1	44.1 ± 1.7	42.8 ± 0.6
EndoG	33.2 ± 0.9	31.0 ± 0.9
ADKIN	81.2 ± 0.2	83.4 ± 0.5
4EIP	85.0 ± 0.7	85.1 ± 0.3

^1^ Mean percentage knock-down compared to NY-SM cells ± standard error of mean.

## Data Availability

Data is contained within the article or Appendix A.

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
