# Peer review of "4E Interacting Protein as a Potential Novel Drug Target for Nucleoside Analogues in Trypanosoma brucei"

_microorganisms, 2021, doi:10.3390/microorganisms9040826_

Round 1
Reviewer 1 Report
Previously, the nucleoside analogue tubercidin has been shown to be effective against Trypanosoma brucei, however with important toxicity against human cells. This manuscript describes the trypanocidal effect and parasite versus human cell selectivity of a several other nucleoside analogues – most of them already previously reported by the authors, one of them newly synthesised – and the further study of one (no 5, with good activity and reasonable SI) to identify the target and mode of action (MoA). Using an RNAi library four candidate targets were identified, from which two were retained after additional susceptibility assays with RNAi clones: adenosine kinase (ADKIN) and 4E-interacting protein (4EIP). However, knockdown of ADKIN did not affect in vitro growth of trypanosomes and normal infectivity in mice was retained, rendering it an unlikely target. In contrast, 4EIP RNAi cells were affected importantly both in vitro growth and mice infectivity. Since 4EIP is known to be involved in differentiation of trypanosomes from proliferating long-slender to quiescent short stumpy forms which are infective for tsetse flies further experiments were performed to assess the effect of compound 5 on this differentiation process and tsetse infectivity. However, no effect was found.
This is well-written manuscript, with the data nicely presented. The experimental strategy followed to determine the target and MoA is elegant and appropriate, and the experiments seem to have been performed competently. Although many questions remain about the MoA of compound 5 (and even more so about any of the other nucleoside analogues mentioned in Figure 1 and Table 1), the work and results described are interesting.
I have only a few minor comments and questions:
- It is not mentioned why specifically compound 5 was chosen for further study, and not one (or more) of the others, for example compound 7 or 8 which have even a slightly better activity and much higher SI.
- Could the authors infer from the results in Figures 5 and 6 that the different nucleoside analogues have distinct targets/MoA? If so, should this not be mentioned?
- Line 45: I think the information about fexinidazole could be stronger than merely mentioning the clinical trial results. The drug has received a positive opinion from the EMA and has been approved for use in the DRC.
- Line 274: The gain for compound 9 is less than 2-fold, that of compound 4 less than 3-fold. Even if the increase in SI is significant, I suggest to phrase the sentence in a somewhat more modest fashion for these two compounds.
- Lines 344-347 and 366-367: Figure 7A-B should be Figure 7A and C, and Figure 7C should be Figure B (although it may be better to swap panels B and C in the figure).
- Line 403: Change ‘were’ to ‘was’.
Author Response
Introduction
We wish to thank you and the reviewers for the very positive and constructive comments and the proposed minor revision. The manuscript as Word-file with the modifications highlighted using “Track changes” is submitted as accompanying file.
Reviewer 1
- It is not mentioned why specifically compound 5 was chosen for further study, and not one (or more) of the others, for example compound 7 or 8 which have even a slightly better activity and much higher SI.
The RNAi experiments were performed on compounds 5 as a follow-up on previously published work where compound 5 came out as lead compound (Hulpia et al. 2019 in the European Journal of Medicinal Chemistry). A continuing synthesis of novel compounds improved the selectivity even more. These compounds were then evaluated on the available RNAi and knockout clones to evaluate the involvement of ADKIN and 4EIP in their mode-of-action.
- Could the authors infer from the results in Figures 5 and 6 that the different nucleoside analogues have distinct targets/MoA? If so, should this not be mentioned?
Indeed, the different degree of dependency on ADKIN and 4EIP of the different nucleoside analogues suggests a different mode-of-action within this group of compounds. A statement is added to the results section.
Line 337-338: The variable dependence of the different compounds on ADKIN and 4EIP suggests a diverse MoA of the nucleoside analogues.
- Line 45: I think the information about fexinidazole could be stronger than merely mentioning the clinical trial results. The drug has received a positive opinion from the EMA and has been approved for use in the DRC.
More information of fexinidazole has been added.
Line 46-48: Fexinidazole has been approved by the European Medicines Agency (EMA) and is approved in the Democratic Republic of Congo (DRC) for the treatment of both stages of HAT.
- Line 274: The gain for compound 9 is less than 2-fold, that of compound 4 less than 3-fold. Even if the increase in SI is significant, I suggest to phrase the sentence in a somewhat more modest fashion for these two compounds.
The sentence was adjusted for compound 4 and 9.
Line 281-282: Analogues 4 and 9 showed a modest increase in selectivity, whereas the remaining analogues showed a significantly improved selectivity.
- Lines 344-347 and 366-367: Figure 7A-B should be Figure 7A and C, and Figure 7C should be Figure B (although it may be better to swap panels B and C in the figure).
Panels B and C of figure 7 have been swapped so that they appear chronologically as mentioned in the text.
- Line 403: Change ‘were’ to ‘was’.
Done (Line 413).
Reviewer 2 Report
The authors proposed a paper entitled “4E Interacting Protein as a Potential Novel Drug Target for 2 Nucleoside Analogues in Trypanosoma brucei” for the publication in Microorganisms, mdpi.
this paper has a quite good scientific soundness and deserves to be published after minor revisions.
the use of English is good and does not require to be revised, in my opinion.
here is a list of some minor issues:
Line 23. what is 5 in this expression “ into the mode-of-action of 5.”? does it refer to the numbered molecules in figure 1? in this case, it should be defined in the abstract section, since it appears in the paper for the first time.
An abbreviation list could be added to this paper according to the journal guidelines
please, improve the quality of molecules representation in the composition of figure 1.
plase, define the aconyms in paragraph 2.4.
provide references referring to the assay proposed and described in paragraph 2.5
check the focus of figure 2a. however, the diagram of figure 2a could be moved to the results section, if the authors agree. or, at least, to supplementary materials.
table 1. please define SEM in this table.
please, better describe and comment diagrams from 3E to 3H
Figure 5. I suggest insert the written text in this figure directly using the program for the figure composition. The successive addition of text results in my opinion into a loss of focus and figure quality.
I suggest developing conclusions with future perspectives.
Author Response
Introduction
We wish to thank you and the reviewers for the very positive and constructive comments and the proposed minor revision. The manuscript as Word-file with the modifications highlighted using “Track changes” is submitted as accompanying file.
Reviewer 2
- Line 23. what is 5 in this expression “ into the mode-of-action of 5.”? does it refer to the numbered molecules in figure 1? in this case, it should be defined in the abstract section, since it appears in the paper for the first time.
The abstract has been changed to avoid using the compound code in this part of the text.
Line 22-23: Whole-genome RNAi screening revealed the involvement of adenosine kinase and 4E interacting protein into the mode-of-action of certain antitrypanosomal nucleoside analogues.
- An abbreviation list could be added to this paper according to the journal guidelines
As indicated in the guidelines, abbreviations are defined the first time they appear in the text and are used consistently thereafter.
- please, improve the quality of molecules representation in the composition of figure 1.
High resolution TIFF files are uploaded separately.
- plase, define the aconyms in paragraph 2.4.
Definitions of the acronyms MEM and MRC5-SV2 are added to paragraph 2.4.
Line 121-122: Human lung fibroblasts (MRC-5SV2 cells) were cultured in Minimum Essential Medium (MEM) (Life Technologies)
- provide references referring to the assay proposed and described in paragraph 2.5
A reference describing the susceptibility assay has been added (Line 137).
Maes, L.; Cos, P.; Croft, S.L. The Relevance of Susceptibility Tests, Breakpoints, and Markers. In Drug Resistance in Leishmania Parasites: Consequences, Molecular Mechanisms and Possible Treatments, Ponte-Sucre, A., Diaz, E., Padrón-Nieves, M., Eds. Springer: Vienna, 2013; pp. 407-429.
- check the focus of figure 2a. however, the diagram of figure 2a could be moved to the results section, if the authors agree. or, at least, to supplementary materials.
Figure 2 has been moved to the supplementary information. High-resolution TIFF files of all the figures are uploaded separately.
- table 1. please define SEM in this table.
Line 294: 1The selectivity index (SI) represents the CC50/IC50. SEM: Standard Error of the Mean.
- please, better describe and comment diagrams from 3E to 3H
Figure 3 has been adjusted to clarify panels E to H.
- Figure 5. I suggest insert the written text in this figure directly using the program for the figure composition. The successive addition of text results in my opinion into a loss of focus and figure quality.
The figure has been adjusted to improve resolution.
- I suggest developing conclusions with future perspectives.
Future perspectives have been added to the conclusion.
Line 460-462: The available RNAi and knockout clones allow a more in depth analysis into the structural requirements for metabolic activation and 4EIP interaction of newly synthetized nucleoside analogues.